# A Multi-Agent Deep Reinforcement Learning-Based Popular Content Distribution Scheme in Vehicular Networks

**DOI:** 10.3390/e25050792

**Published:** 2023-05-12

**Authors:** Wenwei Chen, Xiujie Huang, Quanlong Guan, Shancheng Zhao

**Affiliations:** 1College of Information Science and Technology, Jinan University, Guangzhou 510632, China; chenwenwei@stu2020.jnu.edu.cn (W.C.);; 2Guangdong Key Laboratory of Data Security and Privacy Preserving, Guangzhou 511443, China

**Keywords:** multi-agent deep reinforcement learning (MADRL), popular content distribution (PCD), internet of vehicles, vehicle-to-vehicle (V2V) communications, self-attention, spectral clustering

## Abstract

The Internet of Vehicles (IoV) enables vehicular data services and applications through vehicle-to-everything (V2X) communications. One of the key services provided by IoV is popular content distribution (PCD), which aims to quickly deliver popular content that most vehicles request. However, it is challenging for vehicles to receive the complete popular content from roadside units (RSUs) due to their mobility and the RSUs’ constrained coverage. The collaboration of vehicles via vehicle-to-vehicle (V2V) communications is an effective solution to assist more vehicles to obtain the entire popular content at a lower time cost. To this end, we propose a multi-agent deep reinforcement learning (MADRL)-based popular content distribution scheme in vehicular networks, where each vehicle deploys an MADRL agent that learns to choose the appropriate data transmission policy. To reduce the complexity of the MADRL-based algorithm, a vehicle clustering algorithm based on spectral clustering is provided to divide all vehicles in the V2V phase into groups, so that only vehicles within the same group exchange data. Then the multi-agent proximal policy optimization (MAPPO) algorithm is used to train the agent. We introduce the self-attention mechanism when constructing the neural network for the MADRL to help the agent accurately represent the environment and make decisions. Furthermore, the invalid action masking technique is utilized to prevent the agent from taking invalid actions, accelerating the training process of the agent. Finally, experimental results are shown and a comprehensive comparison is provided, which demonstrates that our MADRL-PCD scheme outperforms both the coalition game-based scheme and the greedy strategy-based scheme, achieving a higher PCD efficiency and a lower transmission delay.

## 1. Introduction

The Internet of Vehicles (IoV) enables data exchange among moving vehicles on the road and fixed RSUs [1,2]. In the past two decades, data dissemination in IoV has become an important research topic because it can improve traffic safety as well as travel comfort [3,4]. There are mainly two types of data for dissemination in IoV: safety-related data and infotainment data. Safety-related data is generally real-time traffic information that should be processed quickly and reliably among vehicles. All vehicles on the road are interested in safety-related data [5]. Infotainment data usually refers to the data that helps to enhance the travel experience and improve travel comfort for passengers [6]. Applications of infotainment data can be classified into two categories: the personalized service that provides tailored content to different vehicles based on specific needs, and the popular content service that delivers data that most vehicles are interested in when located in a specific area or at a specific time. The popular content service requires both low latency and high distribution to ensure quick and efficient delivery to as many passing vehicles as possible. Since safety-related data also requires low latency and high distribution, it can be seen as a special case of popular content. In general, popular content refers to data that is of interest or mandatory for most vehicles, such as popular videos, traffic emergency notifications, and local travel advertisements [7].

In PCD, RSUs distribute popular content to vehicles within their coverage areas, also known as the area of interest (AoI). However, due to the high mobility of vehicles and the limited coverage of RSUs, vehicles may not receive complete content from a single RSU. Moreover, the deployment cost of RSUs is too high to cover all areas, so the number of RSUs available to vehicles is limited. Therefore, collaboration among vehicles via V2V communication is necessary to exchange partial content when they are out of the RSU coverage. However, V2V communication also poses some challenges for PCD. First, wireless communication between vehicles is prone to interference, signal attenuation, and even conflicts. Second, the topology of the vehicular network changes constantly, resulting in unstable wireless links between vehicles.

### 1.1. Related Work

Push-based information dissemination, which can be seen as a special case of PCD, has been investigated. Schwartz et al. [8] proposed the push-based sample and robust dissemination (SRD) protocol by an efficient periodic hello message mechanism that provided limited local topology information for the protocol. The SRD protocol can prevent the problem of broadcast storms in dense networks by an optimized broadcast suppression technique. Di Felice et al. [9] proposed the dynamic backbone-assisted MAC protocol that constructed a virtual backbone of vehicles in IoV and prevented contention using a new forwarding method. Kumar and Lee [10] proposed the P2P cooperative caching scheme, where vehicles request content from neighbours by P2P, while only contacting the RSU if no neighbours could respond. d’Orey et al. [11] proposed a method called neighbour-aware virtual infrastructure, which extended the RSU coverage virtually by selecting some vehicles as a virtual infrastructure based on network load or application requirements. Messages were broadcasted based on these virtual infrastructures. Nawaz Ali et al. [12] proposed the enhanced cooperative load-balancing system to achieve data dissemination through multiple relaying RSUs. However, these works assume small data sizes and do not address the challenges of efficiently delivering large structured content. In addition, the scheme proposed by Nawaz Ali et al. [12] entails a high deployment cost of RSUs.

Delay-tolerant data dissemination, which is an important application in IoV but different from the PCD, has also been usually investigated by adopting clustering and broadcasting approaches. In 2019, Shafi and Venkata Ratnam [13] proposed a cross layer cluster-based routing approach for efficient multimedia data dissemination with improved reliability in IoV. The approach used information from the physical and network layers to dynamically select suitable relay nodes and routing paths to reduce packet loss and delay. In 2022, Shafi and Venkata Ratnam [14] further proposed a different cross layer cluster by the ant colony optimization called ESBCA, for multimedia data dissemination in IoV. The ESBCA approach used different parameters for cluster head selection, such as relative velocity, propagation range, travel time, node density, etc., to improve route stability and data dissemination. Sun et al. [15] proposed a collaborative multicast scheme that combined multicast with D2D-assisted relay for transmitting multimedia data in vehicular networks, and designed a communication quality metric and a relay selection algorithm to optimize data transmission. Jeevitha and Bhuvaneswari [16] proposed a cluster-based cooperative multicast scheme for efficient multimedia data dissemination in hybrid cellular–D2D vehicular networks. The scheme first clustered the multicast group members using cellular signalling, then selected the cooperative forwarding set using graph theory algorithms, and finally used D2D communication to achieve both intra-cluster and inter-cluster multicast. Hasson and Abbas [17] proposed a clustering algorithm, which used a variable range of cluster heads to find cluster heads for each new vehicle. The clustering algorithm helped to make the network more stable with a high delivery rate, high coverage, and low information loss rate. However, these approaches are not suitable for scenarios with long transmission durations due to the dynamic topology of vehicular networks.

Popular content distribution has attracted many researchers due to the fact that popular contents are often not small-scale and are usually of interest to most of the vehicles on the road. One main-stream method for PCD is on the basis of the coalition game theory. In 2013, Wang et al. [18] proposed a PCD scheme based on coalition formation games, in which vehicles were divided into subnetworks, then a greedy algorithm was used to select the best broadcaster in each subnetwork. In 2018, Hu et al. [19] proposed a cooperative transmission scheme based on coalition games, which dynamically clustered vehicles into different groups according to their locations, and then used game theory to find the optimal transmission network structure within each group, so that vehicles could share data blocks and achieve efficient dissemination of popular content. When the vehicle movement caused location changes, the scheme could adaptively update the clusters and transmission network structure. In 2019, Chen et al. [20] proposed a cooperative V2V transmission scheme based on coalition graph games, called CVCG, where a unique utility function was well designed and a dynamic location-based partitioning scheme of vehicles was provided to reduce algorithm complexity and improve the scalability. In 2020, Wang et al. [21] proposed a full-duplex communication scheme based on coalition games, which improved the content transmission efficiency and fairness through cooperation and competition among vehicles, and designed a distributed algorithm that enabled vehicles to autonomously choose suitable coalitions according to their own cache, channel, and demand. In 2022, Zhang et al. [22] proposed an edge vehicle computing collaborative content dissemination strategy based on fuzzy logic and coalition graph games, which used V2V communication to assist RSUs in distributing content, adjusted the relay vehicle ratio according to vehicle density, and improved the content dissemination range and efficiency. However, coalition formation game-based schemes require multiple games between vehicles, inevitably spending a lot of time on reaching some equilibrium.

Different from coalition game-based methods, researchers have used other schemes to improve the efficiency of PCD. Li et al. [7] proposed a cooperative popular content distribution scheme based on the symbol level network coding, called CodeOn, which could effectively improve the content downloading speed and reliability in vehicular networks. In this scheme, the server encoded popular content into symbols and transmitted them to vehicles. Vehicles obtained symbols from the server or other vehicles and stored them in their caches. Vehicles also broadcasted symbols according to their own and their neighbours’ cache statuses for cooperative distribution. Once vehicles had collected enough symbols, they could decode them back into content. Huang and Wang [23] proposed a collaborative downloading scheme for popular content dissemination in urban vehicular networks, which had three innovations: a cell-based clustering strategy, a topology pre-creation and update scheme, and a cross-cluster relay selection and generation selection strategy. Chen et al. [24] proposed a popular content distribution scheme based on the group acknowledgment strategy. The scheme divided popular content into multiple segments and selected appropriate forwarding nodes and retransmission strategies according to the dynamic information of vehicles, to improve data transmission efficiency and reliability in vehicular networks.

With the development of AI technology, machine learning, especially deep reinforcement learning (DRL), has been applied in various fields, which can provide autonomous and adaptive control in vehicular applications [25]. Different reinforcement learning-based data dissemination schemes have been designed for vehicular networks (e.g., see references [26,27,28,29,30]). In 2018, Wu et al. [26] proposed a multi-tier multi-access edge clustering architecture that utilized some mobile vehicles as edges and generated different levels of clusters for integrating multiple types of wireless communication technologies, such as cellular communications, mmWave, and IEEE 802.11p. This clustering was conducted by a fuzzy-logic-based approach, where cluster head nodes were selected (by taking into account the vehicle velocity, network topology, and antenna height) and responsible for providing gateway capabilities between different types of communications. Finally, combined with the Q-learning algorithm, spatial intelligence could be achieved. Similar to [26], in [27], the multi-access vehicular environment with different types of wireless communication technologies was considered, and the idea of choosing some mobile vehicles as edges was also employed to propose a reinforcement learning-based end-edge-cloud collaboration routing scheme to find routes in a proactive manner with a low communication overhead. Mchergui et al. [28] proposed a vehicular network broadcasting scheme based on motion parameters, fuzzy system, and deep learning, which could select the optimal relay nodes according to the relay quality index and disseminate data through a multicast mode. Lolai et al. [29] considered data routing in IoV and used a Q-learning reinforcement algorithm to achieve data packet routing between the source and destination by selecting effective road segments and intermediate vehicles based on various parameters. Lou and Hou [30] used DRL to optimize the Markov decision process model and propose a high-definition map dissemination problem that considers the age of information and degree of interest, then solved it with the proximal policy optimization method. Although there are many achievements of DRL-based data dissemination in vehicular networks, DRL schemes for PCD are still very scarce.

### 1.2. Contributions

In this work, we concentrate on DRL-based vehicular collaborations to assist PCD. However, for multi-vehicle environments, if the traditional DRL is used to control data exchanging behaviours among multiple vehicles, the dimension of the agent’s action will increase exponentially with the number of vehicles. This would cause a large number of invalid actions in the agent’s action space, resulting in a sparse reward problem that affects the training of the agent. To solve the sparse reward problem, we adopt a multi-agent deep reinforcement learning (MADRL) framework. This framework allows each vehicle to correspond to an agent that makes decisions independently, effectively reducing the dimension of the agent’s action. The main contributions of our work are summarized as follows.

A multi-agent deep reinforcement learning-based popular content distribution (MADRL-PCD) scheme in vehicular networks is proposed. We divide PCD into two phases. First, vehicles obtain part of the popular content from a few RSUs in the V2R phase. Then, based on the different parts of the content they own, vehicles form a vehicular ad hoc network (VANET) for data sharing to acquire the whole popular content in the V2V phase. We train the agent to coordinate data transmissions in the VANET by multi-agent reinforcement learning. Simulation results show that our scheme outperforms the greedy-based scheme and the game theory-based CVCG scheme with a lower delivery delay and a higher delivery efficiency.To reduce the complexity of the learning algorithm, we propose a vehicle clustering algorithm based on spectral clustering to divide all vehicles in the V2V stage into multiple non-interfering groups, so that data exchange occurs only among vehicles within the same group. Moreover, by clustering, the number of vehicles within each group does not exceed the number of agents, which is helpful to apply the MADRL efficiently. Based on the vehicle partition, we abstract a multi-vehicle environment training and train a fixed group of intelligent agents using the MAPPO algorithm.When applying MADRL, we define the agent’s observation and the environment’s state as matrix forms. Considering that the agent’s decision is based on the correlation of any two vehicles’ information, we use the self-attention mechanism to construct the agent’s neural network. Moreover, we use the invalid action masking technique to avoid unreasonable actions by the agent and accelerate its training process.

The remainder of this article is structured as follows. Section 2 describes the system model for the PCD process in vehicular networks. The channel model of vehicular communications and the algorithm of vehicle clustering are also introduced, and then the PCD problem is proposed in Section 2. Section 3 introduces the proposed MADRL-PCD scheme in vehicular networks in detail, where the training approach and the MADRL network architecture are presented. Section 4 shows the simulation results of the MADRL-PCD scheme, and provides an exhaustive comparison from different viewpoints. Section 5 concludes our work.

## 2. System Model and Problem Formulation

The system model considered in this work is illustrated in Figure 1. It consists of a straight two-way road equipped with an RSU at each end of the road, where new vehicles enter from both ends of the road and immediately fall within the coverage range of the RSUs. Popular content will be delivered to vehicles passing through this road. Suppose the popular content is divided into *M* small chunks of equal size. In each time slot, the two RSUs randomly broadcast a chunk of the popular content, which may be received by vehicles within their respective coverage ranges. Each vehicle has a memory that caches the chunks of popular content received from the RSU. As the vehicle moves, it eventually leaves the coverage of the RSU. At this moment, the vehicle may have already cached a certain number of content chunks. We denote the cache of vehicle *i* by $C_{i}=c_{i,0},c_{i,1},\ldots ,c_{i,M-1}$ , where $c_{i,j}=1$  indicates that vehicle *i* has cached the *j*-th chunk of the popular content; otherwise $c_{i,j}=0$ .

Once vehicles travel beyond the coverage range of the RSUs, they enter the V2V phase. We assume that vehicles can work in full-duplex mode, allowing them to send and receive data simultaneously. In order to more effectively control and optimize the data exchange process between vehicles in the V2V phase, we use a spectral clustering-based method to classify vehicles before the V2V phase. This classification method divides the vehicles into *B* groups, with each group potentially containing a different number of vehicles. The different groups remain independent of each other and do not interfere with each other. That is, vehicles within each group can exchange data to improve the popular content acquisition rate, but vehicles across groups do not exchange data to avoid communication interference by using different communication channels with different spectra. We denote the set of groups as $B=b_{0},b_{1},\ldots ,b_{B-1}$ .

After clustering, each group of vehicles forms a multi-vehicle environment. In this environment, the vehicles in each group share a wireless channel spectrum. We assign each vehicle a DRL agent and use MADRL to train these agents to collaborate efficiently. Each vehicle deploys a pre-trained MADRL agent and uses the information of all vehicles in the same group to determine its optimal target for receiving data. The vehicles then work together to compose a data exchange strategy for the group, which they follow to send a chunk of popular content to any requesting vehicle. After a fixed transmission time, all vehicles enter the next round of V2V collaboration, which involves re-clustering the vehicles, obtaining a new collaboration strategy, and communicating.

### 2.1. Channel Model

We assume that vehicles and RSUs use dedicated short-range communication in both the V2R and V2V phases. Wireless communication suffers from various fading effects, including large- and small-scale fading. The path loss of large-scale fading is considered and can be characterized by the Friis model [31,32]. The receiving signal power is given as
(1)$$\hat{P_{r}}=\frac{P_{t}G_{t}G_{r}\lambda^{2}}{{\left(4\pi \right)}^{2}d^{2}L_{sys}}$$
where $\hat{P_{r}}$  is the receiving signal power with path loss only, $P_{t}$  is the transmitted power, $G_{t}$  and $G_{r}$  are the transmitter and receiver antenna gain coefficients, respectively, λ  is the electromagnetic wavelength in free space, $L_{sys}$  is the system loss factor, and *d* is the distance between the transmitting and receiving antennas. By introducing the notation $k_{f}≜\frac{G_{t}G_{r}\lambda^{2}}{{\left(4\pi \right)}^{2}d^{2}L}$ , Equation (Equation 1) can be represented as $\hat{P_{r}}≜k_{f}P_{t}$ . The small-scale fading effects can be characterized by the Nakagami-m fading model [33,34]. Due to Nakagami-m fading, the receiving signal power follows a Gamma distribution given by
(2)$$P_{r}∼Gamma\left(m,\frac{\hat{P_{r}}}{m}\right)$$
where $\hat{P_{r}}$  is the power of the receiving signal affected by both pass loss fading and Nakagami fading. The parameter *m* depends on the distance *d* between the sender and receiver. When the sender and receiver are close, *m* is large, and the small-scale fading effect is small. Conversely, when they are far apart, *m* is small and the small-scale fading effect is large, causing the received signal strength to fluctuate more severely. To account for this distance-dependent behaviour, we can express *m* as a piece-wise function [35], given by
(3)$$m=\left\{\begin{array}{cc} 3, & d\leq 50\\ 1.5, & 50<d\leq 150\\ 1, & d>150 \end{array}\right..$$

In the V2R stage, the roadside unit sends data to the vehicles within its coverage area. The signal $Y_{j}$  received by vehicle *j* can be represented as
(4)$$Y_{j}=h_{rsu,j}X_{rsu}+N$$
where $X_{rsu}$  is the transmission signal with power $P_{rsu}$  of the RSU, $h_{rsu,j}$  is the channel gain caused by path loss and multipath effects. *N* is the additive Gaussian white noise with an expectation of 0 and a variance of $\sigma^{2}$ , i.e., $N∼N(0,\sigma^{2})$ . Therefore, the SNR of the vehicle *j* in the V2R phase can be represented as
(5)$$SNR_{rsu,j}=\frac{P_{r}\left(rsu,j\right)}{\sigma^{2}}$$
where $P_{r}\left(rsu,j\right)∼Gamma\left(m,\frac{k_{f}P_{rsu}}{m}\right)$ . The transmission rate in the V2R phase is
(6)$$R_{rsu,j}=\frac{\hat{W}}{2}\log_{2}\left(1+SNR_{rsu,j}\right)$$
where $\hat{W}$  is the total channel bandwidth.

In the V2V stage, vehicles send data to each other. During this stage, the transmission signals between different vehicles will interfere with each other. The received signal of the vehicle *j* can be represented as
(7)$$Y_{j}=h_{i,j}X_{i}+\sum_{k\in \psi_{i}}\left(h_{k,j}X_{k}\right)+N$$
where $h_{i,j}$  and $h_{k,j}$  are the channel gains of the target link from vehicle *i* to vehicle *j* and interfering link from vehicle *k* to vehicle *j*, respectively. $X_{i}$  and $X_{k}$  are the transmission signals of vehicle *i* and vehicle *k*, respectively. $\psi_{i}$  is the set of parallel transmission vehicles interfering with the transmission of vehicle *i*. Therefore, the signal to interference plus noise ratio (SINR) at vehicle *j* in the V2V stage can be represented as
(8)$$SINR_{i,j}=\frac{P_{r}\left(i,j\right)}{\alpha \sum_{k\in \psi_{i}}P_{r}\left(k,j\right)+\sigma^{2}}$$
where $P_{r}\left(l,j\right)∼Gamma\left(m,\frac{k_{f}P_{l}}{m}\right)$ , α  is the interference rate of other transmitting vehicles to vehicle *j*. The transmission rate in the V2V phase is
(9)$$R_{i,j}=\frac{\hat{W}}{2}\log_{2}\left(1+SINR_{i,j}\right).$$

To ensure communication link quality, we set an SINR threshold $SINR_{thre}$ . The receiver can only decode the data correctly if the SINR is higher than this threshold. In other words, the communication link from an RSU or vehicle *i* to vehicle *j* is only valid if $SINR_{j}\geq SINR_{thre}$ .

To avoid interference from the vehicles’ transmitted signals to other vehicles’ receiving signals, we assume that the total wireless channel band is divided into two equal parts. Two neighbouring groups use different bands for communication.

### 2.2. Vehicle Clustering

We apply reinforcement learning to coordinate data exchange among vehicles in the V2V phase. However, this phase may involve a large number of vehicles, which complicates the algorithm’s design. Vehicle clustering is an effective way to enhance IoV communication performance in a large network setting with complicated road conditions and a huge number of vehicles. In most existing studies, most vehicle clustering methods select cluster head nodes first, and then form vehicle groups around these cluster heads. The cluster head node acts as the gateway of the group, and other vehicles in the group primarily communicate with it, thereby realizing communication between different groups. However, our goal of clustering is to limit the communication behaviour between vehicles of different groups by mitigating interferences in V2V collaborations, so there is no need for a complex process of selecting cluster heads. Therefore, we propose a simple and effective vehicle clustering algorithm based on the spectral clustering method [36] to divide vehicles into multiple groups. The vehicle clustering algorithm is a two-stage algorithm. In the first stage, vehicles are partitioned into different groups by spectral clustering. In the second stage, each group is optimized to ensure that the number of vehicles and the inter-vehicle distances are within predefined thresholds.

When applying spectral clustering, we need to determine two important parameters: similarity and the number of clusters. According to Equation (Equation 1), we define the similarity between two vehicles as $\frac{1}{d^{2}}$ , where *d* is the Euclidean distance between them. We denote the position of a vehicle by (x,y) , where *y* corresponds to the lane and *x* is the distance from one of the standard endpoints of the road. This definition reflects the wireless channel quality and the formula is much easier to compute.

The number of clusters, which is also the number of vehicle groups after dividing all vehicles, depends on both vehicle density and inter-vehicle similarity. It can vary under different road environments. Both factors increase the algorithm’s complexity, so we approach it from the perspective of expected results. In the V2V phase, we use MADRL to train agents to control the data exchange behaviour of vehicles within a vehicle group, and the MADRL training process requires a fixed number of agents. This leads to an upper limit on the number of vehicles within a vehicle group, denoted by $N_{b}$ . Therefore, to keep the number of vehicles within each vehicle group no higher than $N_{b}$ , we need to divide the vehicles into at least $\hat{B}=\left\lceil N/N_{b}\right\rceil$  groups. We use $\hat{B}$  as the number of vehicle clusters. According to the definition of similarity, we can construct the similarity matrix *W* and the degree matrix *D*. The *j*-th element $w_{ij}$  in the *i*-th row of the similarity matrix *W* is the similarity $1/d_{ij}^{2}$  based on the distance $d_{ij}$  between vehicle *i* and vehicle *j*. The degree matrix *D* is a diagonal matrix where the element $d_{i}$  in the *i*-th row is the sum of the similarities between vehicle *i* and the other vehicles, i.e., $d_{i}=\sum_{j\neq i}w_{ij}$ . Since spectral clustering converts the clustering problem into a graph optimal partitioning problem through graph Laplacian, we need to construct the graph Laplacian. The unnormalized graph Laplacian can be expressed as
(10)L=D−W.
Then we compute the eigenvectors corresponding to the top $\hat{B}$  smallest eigenvalues of the graph Laplacian *L*, denoted as $u_{0},u_{1},\ldots ,u_{\hat{B}-1}$ . These $\hat{B}$  eigenvectors can form a matrix *U* by columns, denoted as
(11)$$U=\left[\begin{array}{cccc} u_{0} & u_{1} & \ldots & u_{\hat{B}-1} \end{array}\right].$$
Then a new low-dimensional representation *U* of the data points is produced, where each row corresponds to a data point. The data points exhibit more salient clustering features in this representation, which enables us to apply the k-means algorithm for partitioning. By the k-means algorithm, $\hat{B}$  cluster centroids (of vehicles) are randomly initialized, then each vehicle is assigned to the closest cluster based on the Euclidean distance to the centroid. The centroid of each cluster is updated by taking the centre of all the vehicles assigned to that cluster. Then the assignment is repeated and updated until convergence or a maximum number of iterations is reached [37].

However, using only the spectral clustering algorithm may cause two problems as shown in Figure 2. The first problem is that some groups have more than $N_{b}$  vehicles, which exceeds the limit of the trained agents. The second problem is that some groups have vehicles that are too far apart to communicate effectively. To solve these problems, in the second stage, we further process the spectral clustering results. Specifically, we denote a vehicle communication range $r_{veh}$  and assume that vehicles cannot communicate with each other if their distance exceeds $r_{veh}$ . We introduce a queue *Q* and a set $B^{\prime }$ , then put all vehicle groups partitioned by spectral clustering into queue *Q*. Each time we take out the first vehicle group *b* from the queue *Q*, all vehicles in the group are sorted according to their positions and the position of the first vehicle is recorded. Then, we introduce an empty vehicle group $b^{\prime }$ . For each vehicle in group *b*, we first add it to group $b^{\prime }$ . If the number of vehicles in group $b^{\prime }$  exceeds the upper limit $N_{b}$  after adding this vehicle, or if the Eulerian distance between this vehicle and the first vehicle exceeds the communication range $r_{veh}$ , then all other vehicles except it in group $b^{\prime }$  are saved as a legal vehicle group and put into set $B^{\prime }$ , and it is updated as a one-point set of group $b^{\prime }$ , and its position is used as the position of the new first vehicle. Finally, when queue *Q* is empty, set $B^{\prime }$  contains the final clustering results of the vehicles. The overall procedure for the vehicle clustering algorithm is presented in Algorithm 1.

**Algorithm 1** Vehicle clustering algorithm**Input:** All vehicle positions $P=\{p_{0},p_{1},\ldots ,p_{N-1}\}$ , the maximum number of vehicles in a group $N_{b}$ , the communication range $r_{veh}$ **Output:** The partition result $B^{\prime }$ 
1:Calculate the Eulerian distance between all vehicles based on their positions2:Construct a similarity matrix *W* and a degree matrix *D*3:Compute the unnormalized graph Laplacian *L* by Equation (Equation 10)4:Calculate the expected number of vehicle groups $\hat{B}=\left\lceil N/N_{b}\right\rceil$ 5:Compute the first $\hat{B}$  eigenvectors $u_{0},\ldots ,u_{\hat{B}-1}$  of *L* and form them into a matrix $U\in R^{N\times \hat{B}}$  by columns6:Treat each row of *U* as a point in $R^{\hat{B}}$  and cluster them into $\hat{B}$  clusters via k-means clustering algorithm7:Assign each vehicle to the cluster corresponding to its row in *U*, then get the initial partition result B 8:Put all the groups in B  into a queue *Q*9:Set $B^{\prime }=\emptyset$ 10:**while**
 Q≠∅ 
**do**11:     Take out a group *b* from the head of the queue12:     Sort the vehicles in *b* according to a certain coordinate axis13:     Set $b^{\prime }=\emptyset$ 14:     Record the coordinate of the first vehicle $x_{0}$  in *b*15:     **for** 
 i=1,…,∣b∣ 
**do**16:          Add the *i*-th vehicle in *b* to the sub-group $b^{\prime }$ 17:          **if** $|b^{\prime }|>N_{b}$  or $x_{i}-x_{0}>r_{rsu}$  **then**18:               Save all the vehicles in $b^{\prime }$  except the last one as a new group, and add the group to $B^{\prime }$ 19:               Update the sub-group $b^{\prime }$  using the one-point set of the *i*-th vehicle20:               Update the coordinate of the first vehicle $x_{0}=x_{i}$ 21:          **end if**22:     **end for**23:     Add the sub-group $b^{\prime }$  to $B^{\prime }$ 24:
**end while**



### 2.3. Problem Formulation

The main goal of popular content distribution is to reduce the download latency of popular content, i.e., the time for a vehicle to acquire the complete content. We divide the system time *T* into equal-length, non-overlapping time slots. At each time slot *t*, the vehicles are first clustered by the vehicle clustering algorithm proposed in Section 2.2, then the agent forms a collaborative communication policy among the vehicles based on the clustering results, and finally the vehicles transmit data according to the policy. We use $a_{i,j}\left(t\right)\in \left\{0,1\right\}$  to indicate whether vehicle *i* sends a chunk of content data to vehicle *j*. We denote the download latency of vehicle *i* to obtain the popular content by $T_{i}$ , which is the sum of the V2R and V2V phases. We aim to minimize the average time to obtain the complete content for all vehicles, expressed by
(12)$$\min\frac{1}{N}\sum_{i=0}^{N}T_{i}$$
(12a)$$s.t.\sum_{j\in N}a_{i,j}\left(t\right)=1,\forall i\in N$$
(12b)$$\sum_{i\in N}a_{i,j}\left(t\right)=1,\forall j\in N$$
(12c)$$SINR_{i,j}\geq SINR_{thre},\forall i,j\in N$$
where the constraint (12a) ensures that each vehicle only sends data to one target at a time, constraint (12b) ensures that each vehicle only receives data from one neighbour at a time, and constraint (12c) is a necessary condition for the communication link.

## 3. Multi-Agent Deep Reinforcement Learning

We model the collaborative data exchange process as a Markov decision process and use MADRL to control the data transmission of vehicles. MADRL has multiple agents trained in the same environment. These interact with the environment several times, in order to learn changes in the environment as well as each other’s strategies. However, each time the clustering result is different the same agent may interact with different agents in the next time slot. This prevents the effective training of the agents. Therefore, we construct an abstract training environment where each vehicle’s position is fixed. In each training, a random number of vehicles is generated. The number $\hat{N}$  of the vehicles satisfies $3\leq \hat{N}\leq N_{b}$ . The positions of these vehicles are randomly distributed over a section of the road. Each vehicle caches a random number of popular content chunks. Each vehicle is an agent that independently decides whether and from whom to obtain a chunk of content at each slot.

### 3.1. Environmental Modelling

We use decentralized partially observable Markov decision processes (DEC-POMDP) [38] to model our environment. DEC-POMDP consists of the following elements: the agent’s observation of the environment, the agent’s action, the state of the environment, and the reward function. The observation $o_{i}\left(t\right)$  of the agent *i* is the environmental information observed by vehicle *i* at slot *t*, which is used to decide which neighbouring vehicle to request chunks of popular content from. To make a decision, two conditions are required: first, the agent needs to know which neighbouring vehicle has the content chunks that the vehicle *i* needs, i.e., agent needs to know the $C_{i}$  of all neighbouring vehicles; second, the agent needs to know the position (x,y)  of the neighbouring vehicle to check the communication feasibility. Ultimately, we represent the observation $o_{i}\left(t\right)$  in the form of a two-dimensional matrix. The action $a_{i}\left(t\right)$  of agent *i* is the choice of vehicle *i* to request content chunks from at slot *t*. The action $a_{i}\left(t\right)$  is an integer ranging from 0 to $N_{b}-1$ . Since the number of vehicles in the environment may be less than $N_{b}-1$ , and some target vehicles may not provide the required content chunks or be incommunicable, we use the invalid action masking technique [39] to prevent the agent from selecting these invalid targets. It is worth noting that when multiple vehicles choose to request chunks of popular content from the same target vehicle, we adopt a greedy strategy to resolve this conflict. That is, the target vehicle selects the closest vehicle among all requesting vehicles to send content chunks to. The state $s_{i}\left(t\right)$  of the environment consists of both observable and unobservable information of all agents. For simplicity, we use the observable information of all vehicles plus the action of vehicle *i* at the previous slot as the environmental state. The state $s_{i}\left(t\right)$  is also a two-dimensional matrix. For the setting of the reward function, it is obvious that the total time for all vehicles to obtain the whole popular content is reduced when the number of vehicles that receive data successfully at each slot is increased. Therefore, we use the number of vehicles that receive data successfully at slot *t* as the reward signal. To encourage the agents to send popular content chunks as early as possible and actively explore unsampled actions, we introduce the technique of reward shaping. The final reward function is defined as
(13)$$r_{i}\left(t\right)=\beta_{r}N_{done}\left(t\right)+\beta_{p}\left(N_{succ}\left(t\right)-N_{need}\left(t\right)\right)$$
where $N_{done}\left(t\right)$ , $N_{succ}\left(t\right)$ , and $N_{need}\left(t\right)$  are the numbers of vehicles that have acquired the complete popular content, received content chunks, and need to acquire content chunks at time slot *t*, respectively. $\beta_{r}$  and $\beta_{p}$  are the reward factor and the penalty factor, respectively.

Each agent learns an optimal policy to maximize the expected cumulative discounted reward. The cumulative discounted reward of agent *i* taking action $a_{i}\left(t\right)$  at state $o_{i}\left(t\right)$  at a slot *t* is denoted as $Q_{i}\left(o_{i}\left(t\right),a_{i}\left(t\right)\right)$ , and can be described by the Bellman equation as
(14)$$Q_{i}\left(o_{i}\left(t\right),a_{i}\left(t\right)\right)=E\left[r_{i}\left(t\right)+\gamma E\left[Q_{i}\left(o_{i}\left(t+1\right),a_{i}\left(t+1\right)\right)\right]\right]$$
where γ  is the discount factor, $o_{i}\left(t+1\right)$  is the next state of the environment, and $a_{i}\left(t+1\right)$  is next action of agent *i*.

### 3.2. MADRL-PCD Network Architecture

To train the agents, we adopt the multi-agent proximal policy optimization (MAPPO) algorithm. In MAPPO, each agent consists of two neural networks, called the actor network and the critic network. It is worth noting that the critic network is only used in the training phase to help the actor network update and optimize its policy, and in practical use, only the actor network needs to be deployed on the vehicles to make decisions.

The structure of the actor network consists of an input layer, multiple hidden layers, and an output layer as shown in Figure 3a. The input layer is responsible for receiving the observation $o_{i}\left(t\right)$  of the vehicle. The hidden layer mainly consists of the encoder layer of the transformer architecture and the fully connected layer. We use the encoder layer in the hidden layer because we consider that the observation $o_{i}\left(t\right)$  defined in Section 3.1 is a two-dimensional matrix that is suitable as an input to the encoder layer. The most important is that the observation $o_{i}\left(t\right)$  can be considered as a sequence, and each row in $o_{i}\left(t\right)$  is information about a vehicle that can correspond to a data column in the sequence. The relationship between each vehicle’s information and other vehicle information can be captured by the self-attention mechanism in the encoder layer. The fully connected layer is responsible for linearly transforming the output values of the encoder layer to highlight the relationship between the vehicle corresponding to the agent and other vehicles. The output layer takes the output of the hidden layers through a softmax function to obtain a vector of $N_{b}$  elements, represented as the probability distribution of the selection of each neighbouring vehicle by the vehicle corresponding to the agent. To avoid selecting invalid actions, such as selecting vehicles that do not exist or for which no popular content data blocks can be provided, we use an invalid action masking technique before the softmax function by setting the probability value corresponding to the invalid action to zero. After obtaining the probability distribution, if the actor network is actually used, the vehicle corresponding to the subscript with the largest probability value is selected as the best action to target; if it is the training phase, we use a polynomial distribution to randomly sample the actions in order to meet the agent’s exploration of unknown actions.

The structure of the critic network is shown in Figure 3b. Since the environmental state $s_{i}\left(t\right)$  as the input of the critic network is defined similarly to the observation $o_{i}\left(t\right)$ , the structure of the critic network is similar to that of the actor network. However, the output of the final critic network is a predicted value of the cumulative reward for the discount, so the hidden layer of the critic network has one more fully connected layer than the hidden layer of the actor network, which maps the output vector to a value.

## 4. Performance Evaluation

We utilize SUMO (version 1.12.0) [40], a well-established traffic simulation software, to build the simulation environment. The road in the environment is 10,000 m long and each RSU has a coverage of 450 m. The traffic flow follows a Poisson distribution with either 1800 or 3600 vehicles per hour. All vehicles enter from both ends of the road with an average speed of 20 m/s. The speed of each vehicle is perturbed by a normal random variable with a mean of 1 and a variance of 0.1  per time slot. The popular content is divided into multiple chunks, each of 8 MB in size.

The training environment for MADRL consists of a 250 m long road with two lanes in each direction. We set the number of popular content chunks to 16. The actor network consists of 3 transformer encoder layers and 3 fully connected layers, each with 32 neurons. The critic network has a similar structure to the actor network, but includes an additional fully connected layer at the end to convert the output vector into a scalar value. Both networks utilize the leaky ReLU function as their activation function. We use the adaptive moment estimation as the optimization algorithm for both networks, employing learning rates of 0.001  and 0.0005 , respectively. The discount factor γ  is set to 0.9 . The simulation parameters are listed in Table 1.

We compare our proposed MADRL-PCD scheme with the CVCG scheme [20] and a greedy policy-based scheme. The CVCG scheme obtains the vehicle collaboration strategy by establishing a coalition graph game in each group, which is represented by a directed graph. Specifically, starting from an empty graph, the scheme randomly selects a vehicle from the group at each time, and updates the directed graph, which updates the target of sending and receiving data from this vehicle, to improve the vehicle collaboration strategy.This process continues until no choice of vehicle in the group can update the graph. At this point, the directed graph corresponds to the optimal vehicle collaboration strategy for the group. The greedy scheme selects the nearest neighbouring vehicle *j* that can provide the required content chunk to the vehicle *i* among all neighbouring vehicles.

Figure 4 shows the learning process of the MADRL algorithm, where each data point represents the average of 6400 reward values. It can be seen that the average reward obtained by the MADRL algorithm is low at the beginning of the training process. However, as the epoch increases, the average reward gradually increases until it reaches a relatively stable value. The curves indicate that the MADRL algorithm is convergent.

Figure 5 shows a function between the number of time slots and the number of vehicles that receive the entire popular content. The figure also considers popular contents with different sizes of 128 MB and 256 MB, shown in Figure 5a and Figure 5b, respectively. From Figure 5, we can see that the number of time slots increases as the number of vehicles receiving the whole popular content increases. Further, as the size of the popular content grows larger, more V2V collaborations are required, resulting in a larger number of required time slots (i.e., a higher delay). We can also see from Figure 5 that the proposed scheme outperforms both the CVCG scheme and the greedy scheme by taking fewer time slots to achieve a similar delivery efficiency (i.e., the same number of vehicles receiving the whole popular content). The reason is that although the traffic flow remains constant at 1800 vehicles per hour, the vehicle movements and initial content chunks of vehicles entering the V2V phase are identical for all three schemes, more vehicles in the CVCG and greedy schemes cannot complete the popular content acquisition as required. This implies that within the same time (i.e., after experiencing the same number of time slots), the proposed scheme achieves a higher PCD efficiency as more vehicles receive the whole popular content, than the CVCG and greedy schemes, which can also be seen in Figure 5.

For each vehicle, we also count the number of time slots it takes to receive the entire popular content, and then compute the average for the first 100 vehicles. Figure 6 shows the average number of time slots required for each vehicle to obtain the complete popular content at different traffic flows. Figure 6a shows the results for popular content with a size of 128 MB. Our proposed scheme outperforms both the CVCG and greedy schemes. Figure 6b shows the results for popular content with a size of 256 MB. The proposed scheme still outperforms the CVCG scheme, which in turn outperforms the greedy scheme. The advantage of the MADRL-PCD scheme is that fewer vehicles fail to receive the whole popular content within the same time period and under the same conditions as in the CVCG and greedy schemes. Therefore, it requires fewer V2V collaborations to achieve the same content delivery rate as the CVCG and greedy schemes do.

In addition to studying the impact of traffic flow, we investigated how different vehicle speeds affect the time required for vehicles to receive the complete popular content. Figure 7 shows that there is no significant difference in the number of time slots required for each vehicle to obtain popular content with three average vehicle speeds of 10, 20 and 40  m/s. As the vehicle speed decreases, the variation in channel gain between vehicles decreases, resulting in a consistent popular content distribution efficiency at average vehicle speeds of 10 and 20  m/s. On the other hand, as the vehicle speed increases, the channel environment between vehicles changes more frequently, resulting in a higher probability of errors during data transmission and a greater possibility of popular context chunk delivery failure (more time slots are required to obtain the complete popular content). However, since we set a relatively narrow communication distance for vehicles, the popular content distribution efficiency does not decrease significantly even when the average vehicle speed is 40  m/s.

Both the proposed MADRL-PCD scheme and the CVCG scheme rely on a control centre to facilitate vehicle partitioning and environment data collection, as well as communication between vehicles and the centre. Figure 8 shows that the average number of communications for each vehicle when it communicates with the control centre during the target selection phase is a function of the number of time slots. From this figure, we can see that the proposed MADRL-PCD scheme has a lower communication burden than the CVCG scheme. The reason for this is that the proposed MADRL-PCD scheme only involves four steps per time slot: uploading vehicles’ state information, downloading all vehicles’ state information, uploading vehicles’ selected targets, and downloading the final cooperation plan. However, the CVCG scheme, besides the same steps as the proposed scheme, also needs to communicate with the control centre several times to update the directed graph, which can only be performed sequentially.

## 5. Conclusions and Discussion

In this paper, we proposed a multi-agent deep reinforcement learning-based popular content distribution scheme (MADRL-PCD) for delivering the popular content in the IoV. Firstly, we assume that RSUs randomly broadcast popular content chunks, and vehicles acquire parts of the popular content from an RSU when passing through its communication range. Once they leave the RSU’s communication range, vehicles collaborate with each other by exchanging their received popular content chunks. Secondly, we proposed a vehicle clustering algorithm based on spectral clustering to group all vehicles in the V2V phase. This clustering algorithm helps to not only reduce the complexity of the PCD scheme but also improve the overall efficiency of the IoV. Thirdly, we use the MAPPO algorithm to train the neural network. To accelerate the training process of agents, we employed a self-attention mechanism to build the neural network. We also used the invalid action masking technique to prevent the agent from taking invalid actions. Finally, experimental results show that the MADRL-PCD scheme outperforms both the greedy scheme of selecting the nearest vehicle and the game theoretic-based CVCG scheme in terms of delivery efficiency and transmission delay.

In this work, there are some limitations in the system model. One limitation is that each vehicle is assumed to be deployed with a single antenna. Further work may employ the multi-input-multi-output (MIMO) antennas technique to improve PCD by taking into account the MIMO diversity. The other limitation is that each vehicle is assumed to send data to only one target at a time and receivedata from only one neighbour at a time. This restricts the establishment of valid communication links for V2V collaboration. This problem becomes more severe when the number of vehicles in a group is small and there are multiple concurrent transmissions that cause mutual interference. A possible solution is to allow vehicles to broadcast data to multiple targets simultaneously, so that other vehicles in the same group can receive the same data. By choosing appropriate broadcasting vehicles, the intra-group interference can be reduced and the number of valid communication links can be increased, thus enhancing V2V collaboration. Then the efficiency and time cost of PCD could be improved. Therefore, we will focus on designing PCD schemes for broadcast-based V2V communication.

## Figures and Tables

**Figure 1 entropy-25-00792-f001:**
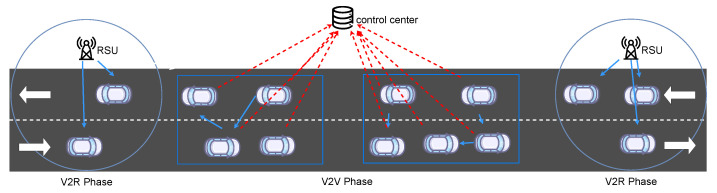
System model.

**Figure 2 entropy-25-00792-f002:**
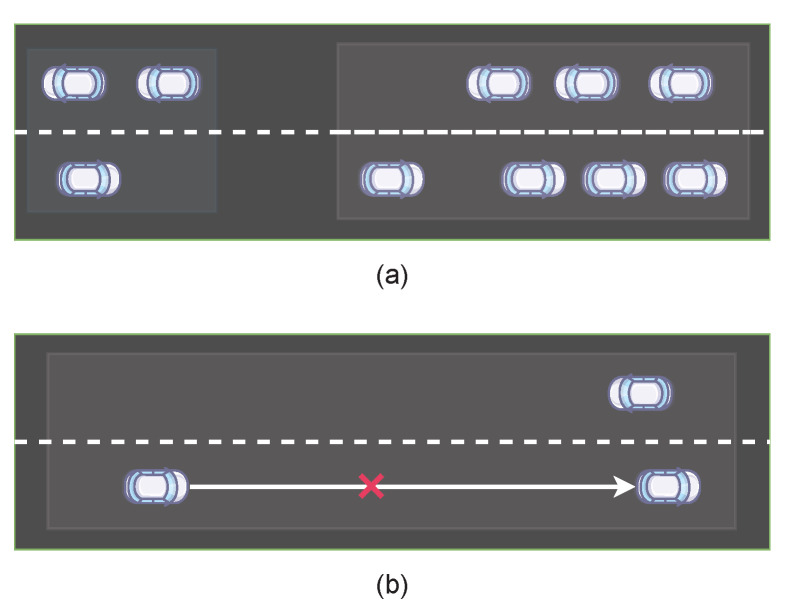
Two cases where the spectral clustering results need to be further segmented. (**a**) A group with too many vehicles, (**b**) A group with no direct communication links between vehicles.

**Figure 3 entropy-25-00792-f003:**
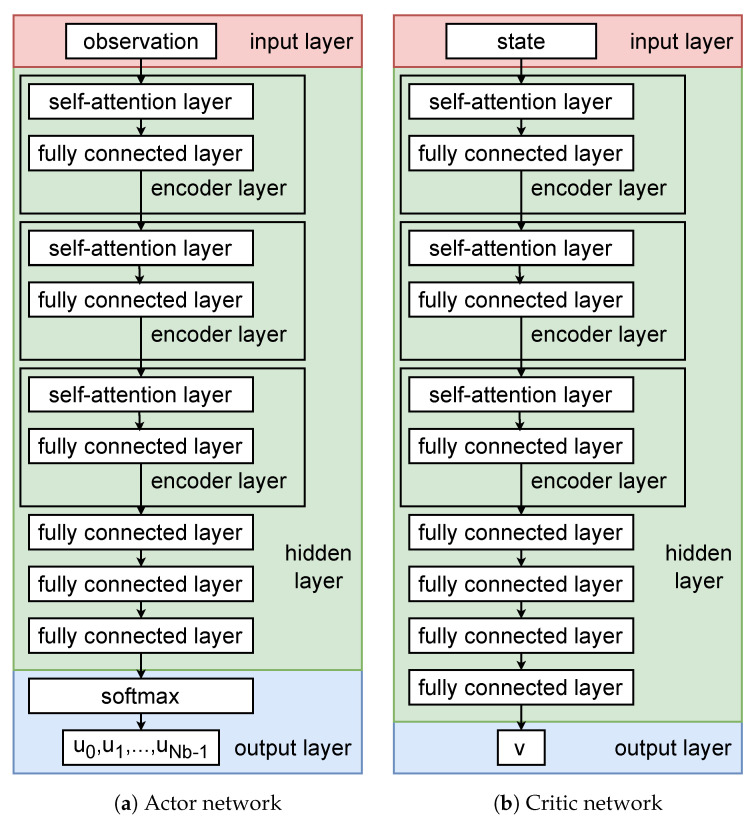
The structure of the actor and critic networks.

**Figure 4 entropy-25-00792-f004:**
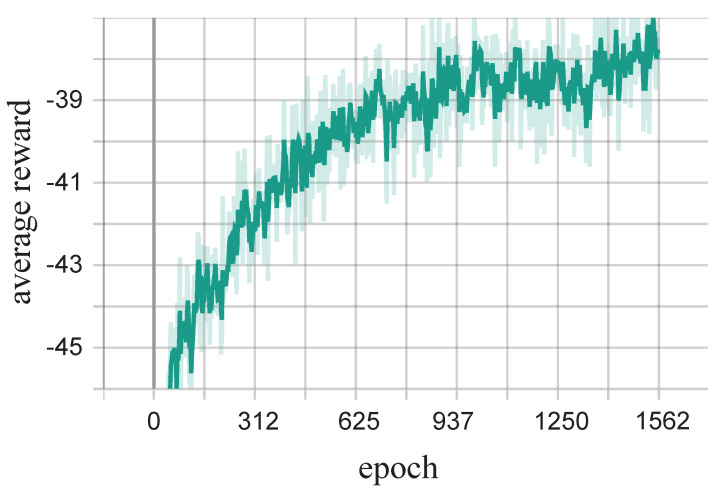
Learning process of the MADRL algorithm.

**Figure 5 entropy-25-00792-f005:**
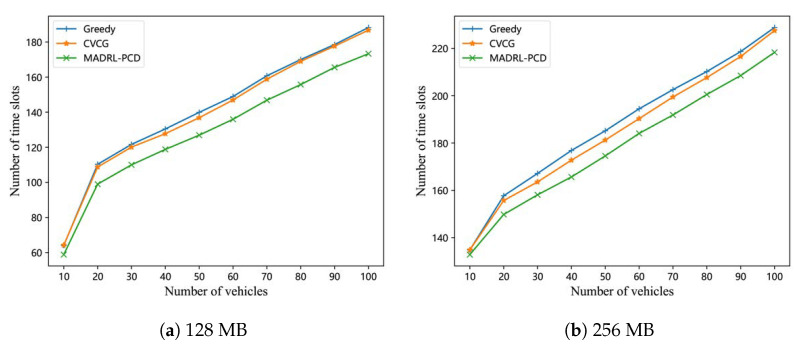
The number of time slots vs. the number of vehicles with the whole popular content.

**Figure 6 entropy-25-00792-f006:**
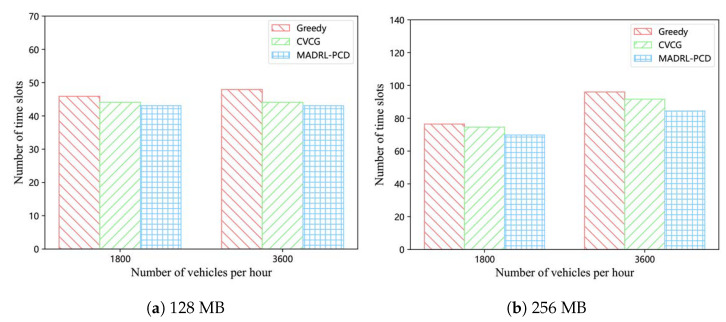
Average number of time slots needed to acquire the whole popular content.

**Figure 7 entropy-25-00792-f007:**
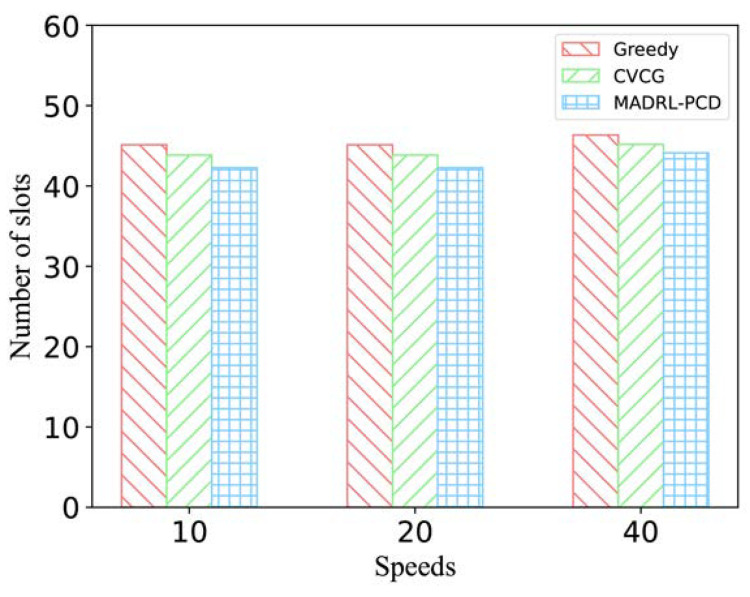
The number of time slots required for vehicles to obtain popular content at different speeds.

**Figure 8 entropy-25-00792-f008:**
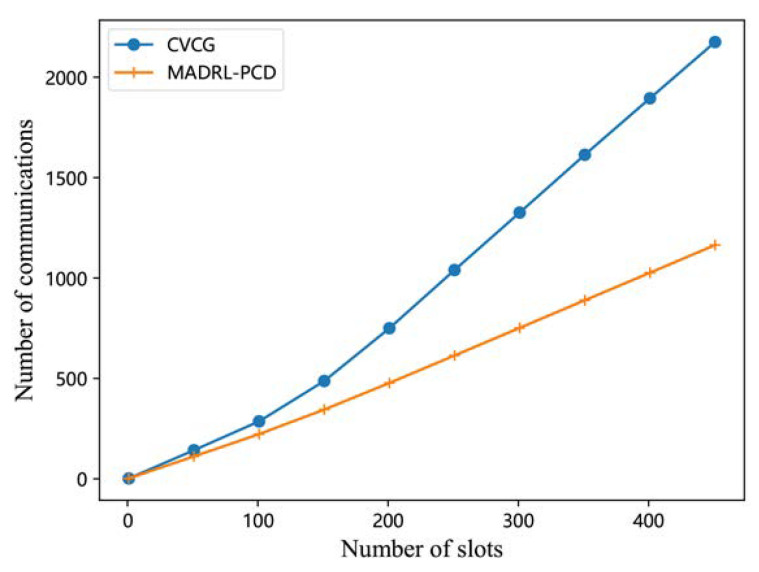
Average number of communications.

**Table 1 entropy-25-00792-t001:** Parameters of simulation.

Parameter	Value
Length of simulation environment	10,000 m
Length of training environment	250 m
Coverage of RSU	450 m
The size of popular content chunk	8 MB
Transmit power of RSU	$P_{rsu}=33$ dBm
Transmit power of vehicles	$P_{veh}=23$ dBm
Antenna gain	$G_{t}=1,G_{r}=1$
System loss	$L_{sys}=1$
Additive Gaussian white noise	$\sigma^{2}=114$ dBm
Interference rate	α=0.1
SINR threshold	$SINR_{thre}=10$ dB
The learning rate of actor network	0.001
The learning rate of critic network	0.0005
Discount factor	γ=0.9
The reward factor	$\beta_{r}=100$
The penalty factor	$\beta_{p}=10$

## Data Availability

Data is contained within the article.

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
