# Peer review of "A Multi-Agent Deep Reinforcement Learning-Based Popular Content Distribution Scheme in Vehicular Networks"

_entropy, 2023, doi:10.3390/e25050792_

Round 1
Reviewer 1 Report
Comments from the reviewer:
1. The authors mentioned MAPPO algorithm in the abstract, the acronym should be explained here.
2. The authors maintained 'due to the high acceptance of the safe-related data, we include it in the popular content category'. The reviewer does not agree the viewpoint. Why the safety-related data are included in the popular content? Shouldn't the popular contents be included in the infotainment?
3 The authors assume the full-duplex operation during V2V phase, do the author consider the difference between half-pulex operation and full-duplex operation, especially regarding the performance, such as BER or PER, as well as SINR?
4 From line 292 to 295, on page seven, the authors describe a series of operations to show how to get clustering results. It should be formulated in math as well.
5 How do the authors simulate/genreate the traffics following the Poisson distribution? What is the simulation tool used here? Why you choose it?
6 How do the authors simulate the V2V and V2R communications? What is the simulation tool used? Why you choose it?
7 By which simulation environment or tool the authors utilized to excecute the learning/traning presented in the paper? How is it connected to your V2V communication
8 Simulation parameters should be listed in a table.
9 The authors assume that 'The popular content is divided into 16 chunks, each with 8 MB', what is the packet size in your simulation? Or a 8 MB chunk data is sent by a vehicle or RSU at a time?
10 In the performance evaluation section, the paper fails to provide overall latency for two communication stages. However, it is the main goal of this paper, as noted in formula (11).
N/A
Reviewer 2 Report
1. The authors mentioned about clustering in vehicular networks. The clustering of vehicular network is a well discussed topic. The authors should explain the novelty of this paper more clearly. There are some recent studies which are strongly related to the clustering-based content distribution, as follows. The authors should discusse these studies in order to explain the scientific problem and contribution more clearly.
https://ieeexplore.ieee.org/abstract/document/9117034
https://ieeexplore.ieee.org/abstract/document/8493113
2. How did you consider the mobility of vehicles in the simulation? Please add more discussions about the effect of vehicle mobility on the performance. Please add some simulation results if possible.
3. There are many typos. A careful revision is required.
There are some typos. A careful revision is required.
Round 2
Reviewer 1 Report
The authors need to check, polish and improve their writing/presentation. There are still a lots of minor issues, for instance:
1) On page 1, line 3, which aims to deliver quickly popular content...
should be 'which aims to quickly deliver popular content'.
2) On page 3, line 112, Zhang et. al proposed a edge vehicle computing...
'a' should be 'an', 'et al' should be in italic style.
and many other similar issues.
3) Next-step research should be directed.
Quality of English Language needs to be improved.
Reviewer 2 Report
No further comments from my side.
No further comments from my side.
Author Response
Dear Reviewer,
Thank you for your valuable comments and suggestions on our manuscript. We appreciate your time and effort in reviewing our work.
We have proofread and edited our manuscript thoroughly to improve the clarity and readability of the English language. We hope that these changes have addressed your concern.
We thank you again for your constructive feedback and hope that you will find our revised manuscript suitable for publication.
Sincerely,
Wenwei Chen